# TMEM16A chloride channel does not drive mucus production

Filipa B Simões, Margarida C Quaresma, Luka A Clarke, Iris AL Silva, Ines Pankonien, Violeta Railean, Arthur Kmit, Margarida D Amaral

**Airway mucus obstruction is the main cause of morbidity in cystic fibrosis, a disease caused by mutations in the CFTR Cl⁻ channel. Activation of non-CFTR Cl⁻ channels such as TMEM16A can likely compensate for defective CFTR. However, TMEM16A was recently described as a key driver in mucus production/secretion. Here, we have examined whether indeed there is a causal relationship between TMEM16A and MUC5AC production, the main component of respiratory mucus. Our data show that TMEM16A and MUC5AC are inversely correlated during differentiation of human airway cells. Furthermore, we show for the first time that the IL-4–induced TMEM16A up-regulation is proliferation-dependent, which is supported by the correlation found between TMEM16A and Ki-67 proliferation marker during wound healing. Consistently, the notch signaling activator DLL4 increases MUC5AC levels without inducing changes neither in TMEM16A nor in Ki-67 expression. Moreover, TMEM16A inhibition decreased airway surface liquid height. Altogether, our findings demonstrate that up-regulation of TMEM16A and MUC5AC is only circumstantial under cell proliferation, but with no causal relationship between them. Thus, although essential for airway hydration, TMEM16A is not required for MUC5AC production.**

## Introduction

Mucus clearance or mucociliary transport (MCT) consists the co-ordinated integration of ion transport, water flow, mucin secretion, cilia action, and coughing, resulting in the continuous flow of fluid and mucus on airway surfaces (Button et al, 2008). Mucus is, thus, an efficient system for protecting the epithelium from the dele-terious effects of inhaled pollutants, allergens, and pathogens, namely, bacteria, by promoting their clearance and separating them from the epithelial cells, thereby inhibiting inflammation and infection (Hansson, 2012; Roy et al, 2014). Mucus is a gel formed by 97% water and 3% solids (mucins, non-mucin proteins, ions, lipids, and antimicrobial peptides) (Fahy & Dickey, 2010). Mucins are

heavily (2–20 × 10⁵ kD) glycosylated proteins (50–90% glycan content) that constitute the main structural components of mucus (1%). The main mucins present in human airway mucus are MUC5AC and MUC5B, which are mostly secreted from goblet cells at the surface airway epithelium and by submucosal glands, respectively (Buisine et al, 1999; Bonser & Erle, 2017).

Mucus hyperproduction and mucociliary dysfunction are major features of many airway obstructive pulmonary diseases, such as chronic obstructive pulmonary disease, asthma, and cystic fibrosis (CF) (Adler et al, 2013). Specific inflammatory/immune response mediators lead to mucus hyperproduction in each of these chronic airway diseases through activation of mucin gene expression and airway remodeling, including goblet cell metaplasia or hyperplasia (GCM/H: reviewed in (Rose & Voynow, 2006)). Whereas metaplasia implies a change in the phenotype of a fully differentiated cell, hyperplasia is caused by cell proliferation (Williams et al, 2006). Importantly, mucin overproduction and GCM/H, although linked, are not synonymous and may result from different signaling and gene regulatory pathways (Rose & Voynow, 2006).

CF, also known as mucoviscidosis, is a disease with major re-spiratory involvement characterized by clogging of the airways by a highly viscous mucus (Ehre et al, 2014), which is its most prominent hallmark and cause of morbidity and mortality (Boucher, 2007). This genetic condition is caused by mutations in CFTR, a cAMP-gated chloride (Cl⁻) and bicarbonate (HCO₃⁻) channel expressed at the apical membrane of epithelial cells in different tissues, including the airways (Kreda et al, 2012). CFTR is also a negative regulator of the epithelial Na⁺ channel (ENaC) (König et al, 2001). As these ions are essential to drive the water flow, CF patients have a dehydrated airway surface liquid (ASL) and reduced water content in mucus (Matsui et al, 2006), impaired MCT, and extensive mucus plugging (Boucher, 2007). This is further exacerbated because of CFTR per-meability to HCO₃⁻, which is essential in the extracellular space for proper mucus release, probably by promoting Ca²⁺ and H⁺ exchange from the mucin-containing intracellular granules, thus facilitating mucin expansion (Garcia et al, 2009; Gustafsson et al, 2012).

Individuals with CF not only have mucus plugging in the airways (and in the ducts of several organs) but also mucus stasis. This has been proposed to result from dehydration of both ASL and mucus

University of Lisboa, Faculty of Sciences, BioISI–Biosystems & Integrative Sciences Institute, Lisboa, Portugal

Correspondence: mdamaral@fc.ul.pt

leading to abnormally high mucus viscosity and deficient MCT (Kreda et al, 2012). Nevertheless, according to other authors, impaired MCT in CF is not due to ASL depletion, but rather to the fact that secreted mucus strands remain tethered to submucosal gland ducts (Hoegger et al, 2014). Moreover, it was shown that inhibition of anion secretion in non-CF airways replicates these CF abnormalities (Hoegger et al, 2014). More recently, based on data obtained in newborn CFTR knockout piglets, it was proposed that MUC5AC (produced by goblet cells) anchors the mucus bundles, mostly composed by MUC5B (produced by submucosal glands), thus being the key controller of mucus transport (Ermund et al, 2017; Xie et al, 2018). Furthermore, the number of MUC5AC-mediated anchorage points in CF mucus is much higher than in non-CF mucus, and without sufficient $HCO_3^-$, the mucus cannot detach from its goblet cell anchor, initiating CF lung disease (Ermund et al, 2017; Xie et al, 2018). Altogether, these data indicate that MUC5AC is the key responsible for mucus stasis in CF.

Clearance of these secretions is a major objective of CF care, typically involving daily chest physiotherapy (Castellani, 2018). Notwithstanding, measurements performed in individuals with CF during stable CF disease found that the vol/vol quantity of MUC5AC protein was ~90% less than in normal mucus, and the mucin-associated sugars were about half of those present in non-CF mucus. However, during exacerbations, levels of MUC5AC protein significantly increased by ninefold in comparison with periods of stable disease in the same individual. Levels of MUC5B also increased, but far less than MUC5AC (Henke et al, 2004).

One long-sought way to compensate for the absence of functional CFTR and thus benefit individuals with CF has been the activation of non-CFTR Cl⁻ channels (Verkman & Galietta, 2009; Mall & Galietta, 2015; Li et al, 2017). Among possible candidates, transmembrane protein 16 A (TMEM16A), also known as anoctamin 1 (ANO1), stands out (Pedemonte & Galietta, 2014; Sondo et al, 2014; Mall & Galietta, 2015), a $Ca^{2+}$-activated Cl⁻ channel (CaCC) which is expressed at the apical membrane of airway epithelial cells (Huang et al, 2009; Scudieri et al, 2012). Indeed, its higher expression levels in goblet cells suggest that it is important for the release and hydration of mucins and, thus, may circumvent the primary defect in CF (Sondo et al, 2014). TMEM16A expression is controlled by pro-inflammatory stimuli, namely, by the Th 2 cytokines IL-4 and IL-13 (Huang et al, 2012; Scudieri et al, 2012; Lin et al, 2015; Kang et al, 2017) and is shown to be induced by asthma-like conditions, that is, in ovalbumin-challenged mice (Benedetto et al, 2019; Huang et al, 2012) , in pig airway tissues treated with histamine (Kang et al, 2017) and in biopsies from asthmatic patients (Huang et al, 2012). Moreover, recent reports have suggested that TMEM16A plays a critical positive role in mucus production/secretion (Huang et al, 2012; Scudieri et al, 2012; Lin et al, 2015; Qin et al, 2016; Kang et al, 2017; Kondo et al, 2017; Benedetto et al, 2019). Indeed, when up-regulated, TMEM16A was reported to co-localize with MUC5AC at the apical membrane of goblet cells (Huang et al, 2012; Scudieri et al, 2012). Should TMEM16A stimulation indeed cause MUC5AC hyper-production, applying such a treatment in CF would further enhance a major symptom in this condition, with a harmful instead of a beneficial result. Nonetheless, the mere fact that TMEM16A and MUC5AC are co-activated by Th-2 cytokine stimulation does not imply a causal relationship between them. So, the true role of TMEM16A up-regulation in mucus production/secretion is still unclear, but its

elucidation is of elevated importance. Moreover, pathways occurring in asthma have been shown to significantly diverge from those in CF (Clarke et al, 2015).

Our goal here was, thus, to examine whether there is a causal relationship between up-regulation of TMEM16A and mucus production, namely, MUC5AC, as previously suggested. To that end, we used a recently described human respiratory basal cell line (BCi-NS1.1) that differentiates into the various respiratory cell types (Walters et al, 2013). Our data show that there is an inverse correlation between TMEM16A and MUC5AC expression levels during differentiation of these multipotent basal cells into different human airway cell types. Indeed, at early differentiation stages, when cells are still proliferating, TMEM16A levels are high and MUC5AC are low, and as cells differentiate into various cell types, the opposite is observed. Moreover, TMEM16A and MUC5AC also have distinct spatiotemporal localizations in these differentiated cell types. Furthermore, we show that under stimulation with IL-4 (a strong inducer of both TMEM16A and MUC5AC) TMEM16A is only up-regulated when proliferation of differentiated cells occurs. Indeed, IL-4 stimulation in the presence of a proliferation blocker no longer induces TMEM16A, in contrast to MUC5AC levels, which still go up. These results demonstrate that MUC5AC production is independent of TMEM16A. In addition, we find that during wound healing, expression levels of TMEM16A rise concomitantly with the proliferation marker Ki-67 further corroborating that proliferation triggers TMEM16A up-regulation.

Altogether, our findings clearly show for the first time in human airways that TMEM16A up-regulation by IL-4 is proliferation-dependent and that this channel is not essential for MUC5AC production, thus remaining a good target for activation in CF and likely other obstructive airway diseases.

# Results

### Analysis of TMEM16A and MUC5AC during differentiation of airway basal epithelial cells

An association between the expression of TMEM16A and MUC5AC was previously reported by several authors (Huang et al, 2012; Scudieri et al, 2012; Lin et al, 2015; Qin et al, 2016; Kang et al, 2017; Kondo et al, 2017; Benedetto et al, 2019). We, thus, first examined the expression levels of these two proteins during differentiation of BCi-NS1.1 cells. To this end, protein was collected from cells every 5 d during a 30-d period of differentiation in air–liquid interface (ALI) culture, and TMEM16A and MUC5AC were detected by Western blot (WB) with respective antibodies (see the Materials and Methods section). In parallel, RNA was also extracted at three different time points (0, 15, and 30 d). Our data clearly show that TMEM16A expression decreases during differentiation, at both transcript and protein levels (Fig 1A–C). In contrast, this decrease in TMEM16A levels occurs simultaneously with an increase in the expression of cell type–specific markers and transepithelial electrical resistance (TEER) (Fig S1), including MUC5AC, which was found to increase significantly during differentiation (Fig 1D–F). These data show an inverse correlation between TMEM16A and MUC5AC expression during differentiation (Fig 1G).

Next, we assessed by immunofluorescence the cellular localization of TMEM16A and MUC5AC in non-differentiated (day 0) and

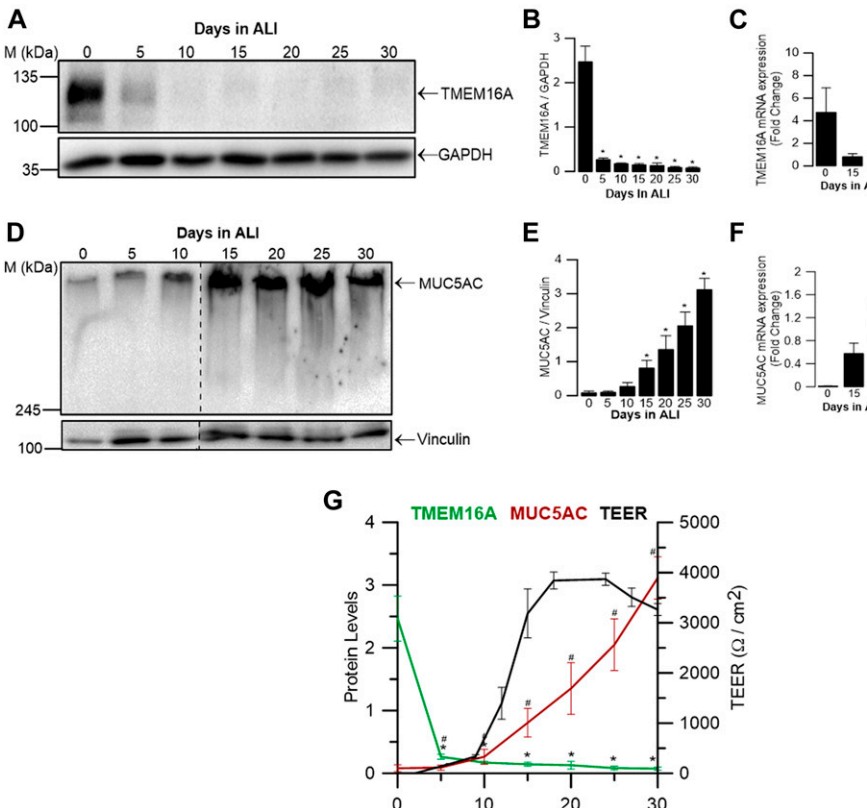

**Figure 1.  TMEM16A and MUC5AC expression levels are inversely correlated during differentiation of BCi-NS1.1 cells.**
**(A)** Time course levels of endogenous expression of TMEM16A protein during differentiation of BCi-NS1.1 cells grown at ALI (days 1–30) detected by WB, showing both its non-glycosylated (~100 kD) and glycosylated forms (~120 kD). GAPDH (~36 kD) was used as a loading control. **(A, B)** Quantification by densitometry of TMEM16A total protein levels from (A) normalized to the loading control shown as mean ± SEM (n = 3). **(C)** Fold-change in TMEM16A mRNA expression levels as a time course of differentiation of BCi-NS1.1 cells grown at ALI (days 0, 15, and 30), determined by qRT-PCR. Fold-change values are mean ± SEM, relative to the mean value of day 30 (n = 3). **(D)** Time course levels of endogenous expression of MUC5AC (>300 kD) analysed by WB during differentiation of BCi-NS1.1 cells. Vinculin was used as a loading control (~120 kD). Dashed line indicates lanes run on the same gel but noncontiguous. **(D, E)** Quantification by densitometry of MUC5AC protein levels from (D) normalized to the loading control shown as mean ± SEM (n = 4). **(F)** Fold-change in MUC5AC mRNA expression levels as a time course of differentiation of BCi-NS1.1 cells grown at ALI (days 0, 15, and 30), determined by qRT-PCR. Fold-change values are mean ± SEM, relative to the mean value of day 30 (n = 4). **(G)** Correlation of TMEM16A and MUC5AC normalized protein levels during differentiation and TEER measurements. Asterisks and cardinals indicate significant difference compared with day 0 (P-value < 0.05, unpaired t test). Source data are available for this figure.

differentiated (day 30) BCi-NS1.1 cells by confocal microscopy (Fig 2). Consistent with WB and qRT-PCR results, our data show a decrease in TMEM16A and an increase in MUC5AC-staining densities between day 0 and 30. Furthermore, TMEM16A localization changes as cells differentiate; although it is present in all non-differentiated (basal) cells with a predominant intracellular distribution at day 0, it is only expressed in some cells and apically localized at day 30 (Fig 2A and B). To distinguish the apical from the basolateral membrane, we stained the apical membrane with an antibody against the tight junction barrier protein ZO-1 (Fig 2A and B). In contrast, MUC5AC is not present at day 0 but is apically secreted at day 30 and only from specific cells (Fig 2C and D).

### TMEM16A is up-regulated in proliferating cells

Our next goal was to further explore this inverse correlation found between TMEM16A and MUC5AC during differentiation of BCi-NS1.1 cells (Figs 1 and 2). Notably, TMEM16A has for long been associated with cell proliferation and tumour growth, being up-regulated in several cancer types (Wanitchakool et al, 2014; Jia et al, 2015; Wang et al, 2017). Therefore, we hypothesized that the increased expression levels of TMEM16A observed at the early stages of differentiation could be related to cell proliferation.

We, thus, determined by WB the expression levels of the proliferation marker Ki-67 as a time course of differentiation of BCi-NS1.1 cells grown at ALI (days 1–30) and observed, as expected, its significant decrease during differentiation (Fig 3A and B), paralleling that of

TMEM16A (Figs 1A–C and 3C). These data, thus, support the concept that high levels of TMEM16A expression correlate with cell proliferation.

### TMEM16A is induced by wound healing in differentiated cells in BCi-NS1.1 cells

To further confirm this hypothesis, monolayers of differentiated BCi-NS1.1 cells were subjected to a wound healing experiment (by scratching) on ALI day 20, that is, a time point when expression levels of both TMEM16A (Fig 1A–C) and Ki-67 are low (Fig 3A and B). During the healing process, the levels of these two proteins were determined by WB at 8, 24, 48 h, and 5 d after injury. Wound healing results indicate that the expression levels of both TMEM16A and Ki-67 increase dramatically after wounding and decrease significantly during the healing process (Fig 3D and E). These results further support the association between TMEM16A induction and cell proliferation.

### BCi-NS1.1 results are confirmed in primary cultures of human bronchial epithelial (HBE) cells

Because BCi-NS1.1 is an immortalized cell line derived from the bronchi of a healthy subject, we next aimed to validate the above data in primary cultures of HBE cells. Remarkably, we also found that TMEM16A and Ki-67 protein expression decrease during differentiation of HBE cells (Fig 4A and B), whereas MUC5AC levels increase (Fig 4C and D), confirming the experiments performed with

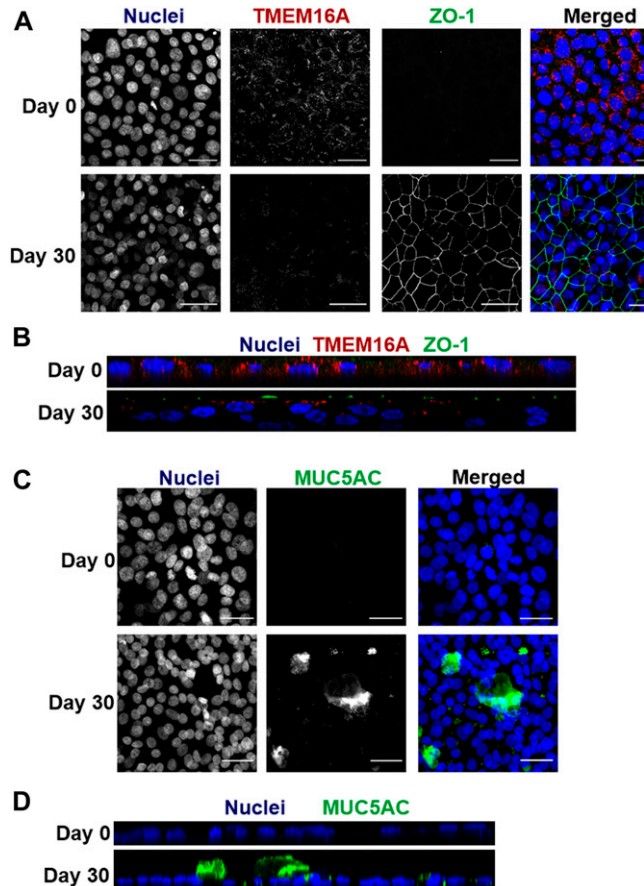

**Figure 2. TMEM16A and MUC5AC have distinct spatiotemporal localizations in differentiating BCi-NS1.1 cells.**

**(A)** Confocal immunofluorescence microscopy images showing TMEM16A localization in permeabilised BCi-NS1.1 cells on days 0 and 30 (upper and lower rows, respectively) of differentiation. Left panels: Nuclei stained with Hoechst stain. Middle panels: endogenous TMEM16A, detected by Alexa Fluor 568 fluorescence. Endogenous ZO-1 was detected by Alexa Fluor 488 fluorescence. Right panels: merged image of the three fluorescent channels: Blue, Alexa 647; Red, Alexa 568; and Green, Alexa 488. Images were acquired with a Leica TCS SP8 confocal microscope (objective 63× oil, NA 1.4). Scale bar = 30 $\mu$m. (n = 3). **(B)** Z-stack of a representative group of cells showing TMEM16A and ZO-1 staining on days 0 and 30 (upper and lower panels, respectively). **(C)** Confocal immunofluorescence microscopy images showing MUC5AC expression in permeabilised BCi-NS1.1 cells on days 0 and 30 of differentiation (upper and lower rows, respectively). Left panels: nuclei stained with metal green, represented by Alexa Fluor 647 fluorescence. Middle panels: endogenous MUC5AC, detected by Alexa Fluor 488 fluorescence. Right panels: merged image of the two fluorescent channels: Green–Alexa 488; Blue–Hoechst. Images were acquired with a Leica TCS SP8 confocal microscope (objective 63× oil, NA 1.4). Scale bar = 30 $\mu$m. (n = 3). **(D)** Z-stack of a representative group of cells showing MUC5AC stained apically on days 0 and 30 (upper and lower panels, respectively).

BCi-NS1.1 cells and indicating that this cellular system is robust and recapitulates the physiological properties of human airways.

### Up-regulation of TMEM16A induced by IL-4 is driven by cell proliferation

Altogether, the above expression and localization data on TMEM16A and MUC5AC strongly suggest that MUC5AC production is independent

of TMEM16A, not just in BCi-NS1.1 cells but also in primary HBE cells. Nevertheless, these data appear contradictory to previous reports suggesting that TMEM16A plays a key role in mucus production, namely, in asthma where both TMEM16A and MUC5AC are concomitantly up-regulated by IL-4 or IL-13 induction (Huang et al, 2012; Scudieri et al, 2012; Lin et al, 2015; Kang et al, 2017).

Thus, given that TMEM16A is up-regulated under proliferation (Fig 3), we next investigated whether the conditions previously shown to significantly induce TMEM16A expression levels, such as the pro-inflammatory cytokine IL-4 (Caputo et al, 2008), also affect cell proliferation. We, thus, postulated that perhaps IL-4 also triggers cell proliferation. Because at day 30 (at ALI) in fully differentiated BCi-NS1.1 cells, TMEM16A expression levels were already undetectable by WB (Fig 1A and B), we chose this time point to test IL-4 induction. As expected, upon treatment with IL-4 for 48 h, a significant up-regulation of this channel occurred (Fig 5A and B, left panels). Consistently, Ussing chamber experiments show that this cytokine also enhances the ATP-activated currents, which are inhibited by the TMEM16A blocker, CaCC-AO1 (Fig 5C and D). Moreover, after removing ATP and CaCC-AO1 solution, a third application of ATP was still effective (Fig S2). Concomitantly, and also as previously described (Temann et al, 1997; Scudieri et al, 2012; Gorrieri et al, 2016), MUC5AC levels also increased upon IL-4 stimulation (Fig 5A and B, middle panels). Notwithstanding, and confirming our hypothesis, IL-4 also led to a significant increase in Ki-67 expression in these fully differentiated cells, thus indicating that cells underwent proliferation (Fig 5A and B, right panels).

To understand whether proliferation drives TMEM16A up-regulation (or the other way around), we performed IL-4 stimulation in the presence of the proliferation blocker mitomycin C and then determined protein levels of Ki-67 and TMEM16A by WB (Fig 5E and F). Remarkably, our data clearly show that under blockage of cell proliferation, TMEM16A is not up-regulated, despite the presence of IL-4, concomitantly with the expected lack of increase in Ki-67 expression levels (Fig 5E and F). As a control, we could observe that the expression levels of the Na$^+$/K$^+$ ATPase pump remain constant (Fig 5E and F). These data strongly favour that the driver of TMEM16A up-regulation is cell proliferation. But curiously, under proliferation arrest, the expression levels of MUC5AC still go up upon IL-4 stimulation (Fig 5G and H), as previously reported for this cytokine (Temann et al, 1997; Scudieri et al, 2012; Gorrieri et al, 2016). These data further support that MUC5AC production is not dependent on TMEM16A.

### TMEM16A is not essential for mucus production in BCi-NS1.1 cells

Increasing evidence accumulated in support of MUC5AC and TMEM16A being uncoupled. Thus, our next goal was to determine that indeed there is no causal relationship between TMEM16A and MUC5AC production/secretion. To this end, we looked into conditions which, despite leading to mucus hypersecretion, do not lead to hyperplasia (i.e., no cell proliferation) unlike IL-4/IL-13 stimulation. One such situation that positively regulates mucus hypersecretion is stimulation by activators of the notch signaling pathway, such as DLL4 (delta-like ligand 4), which directs epithelial differentiation into secretory cells by metaplasia, but no hyperplasia, that is, without proliferation (Guseh et al, 2009; Rock & Hogan, 2011). Strikingly, our results obtained in BCi-NS1.1 cells treated (at day 0 of

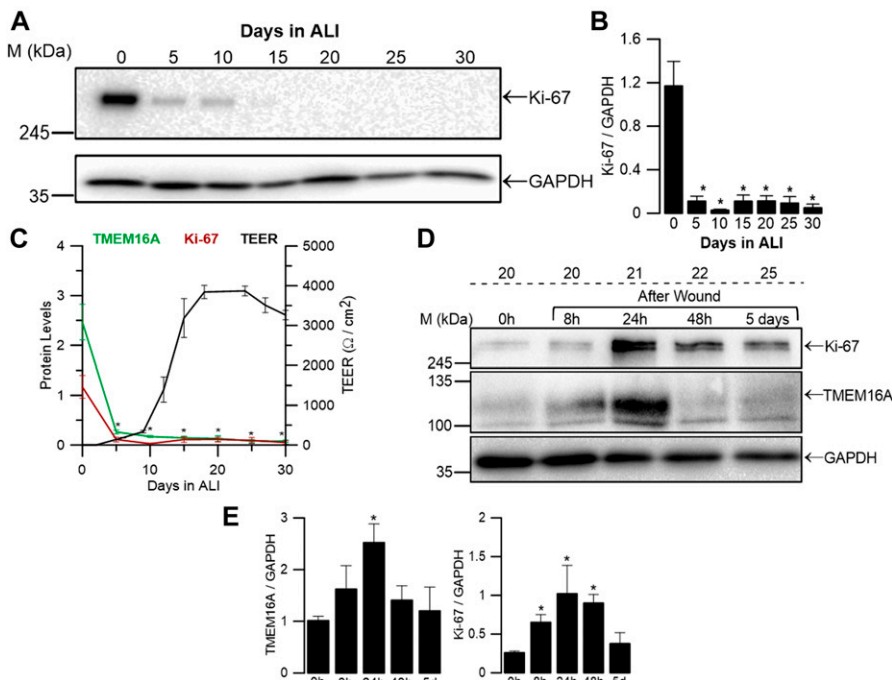

**Figure 3. High TMEM16A expression levels correlate with cell proliferation in BCi-NS1.1 cells after wound healing.**
**(A)** Time course levels of endogenous expression of Ki-67 protein (~300 kD) during differentiation of BCi-NS1.1 grown at ALI (days 0-30) analysed by WB. GAPDH (~36 kD) was used as loading control. **(B)** Quantification by densitometry of Ki-67 expression detected by WB and normalized to the loading control shown as mean ± SEM (n = 3). **(C)** Correlation of TMEM16A and Ki-67 normalized protein levels and TEER measurements during differentiation. Summary of TMEM16A total protein levels represents data in Fig 1. **(D)** Time course (0 h to 5 d) of expression levels of endogenous expression of TMEM16A and Ki-67 during wound closure (0, 8, 24, 48 h, and 5 d after injury) on the 20th day of differentiation at ALI analysed by WB. GAPDH (~36 kD) was used as loading control. **(E)** Quantification by densitometry of total TMEM16A and Ki-67 expression detected by WB and normalized to the loading control shown as mean ± SEM (n = 3–5). Asterisks indicate significant difference compared with day 0 (in graph (B)) or before wound (in graph (E)) (*P*-value < 0.05, unpaired *t* test).
Source data are available for this figure.

ALI) with DLL4 for 30 d did not show any elevation in TMEM16A levels, despite the significant increase in MUC5AC levels (Fig 6). These data strongly suggest that the concomitant rise of TMEM16A and MUC5AC levels under IL4/IL-13 stimulation reported by many groups (Huang et al, 2012; Scudieri et al, 2012; Lin et al, 2015; Qin et al, 2016; Kondo et al, 2017) is coincidental but not causal; whereas TMEM16A is up-regulated because of goblet cell *hyperplasia* (i.e., proliferation), that of MUC5AC results from the higher number of *mucus producing cells*.

### Effect of regulating TMEM16A activity on ASL height

Because our data so far have uncoupled TMEM16A from a possible causal effect on mucus, this channel seems to remain a good potential drug target for CF through its activation (not inhibition) to compensate for the absence of CFTR-mediated $Cl^-/HCO_3^-$ secretion. So, next we aimed to observe the effects of inhibiting this channel on ASL height. Indeed, by inhibiting TMEM16A with the specific inhibitor Ani9 (Seo et al, 2016), we observed a decrease in ASL height compared with control cultures which was significant at all time points (Fig 7). Thus, inhibiting TMEM16A causes significant airway dehydration by reducing fluid secretion, and TMEM16A potentiation remains a good target for hydrating CF airways.

## Discussion

Mucus hyperproduction is a feature that characterizes nearly all airway obstructive pulmonary diseases, including CF, asthma, and chronic obstructive pulmonary disease (Williams et al, 2006; Adler et al, 2013). Therefore, understanding the molecular mechanisms

behind mucus synthesis and/or release is key to developing disease-specific therapies (Rogers & Barnes, 2006). In particular, individuals with CF suffer from extensive mucus plugging and stasis, resulting from airway dehydration and mucus hyperproduction, which leads to impaired MCT and breathing due to malfunctioning

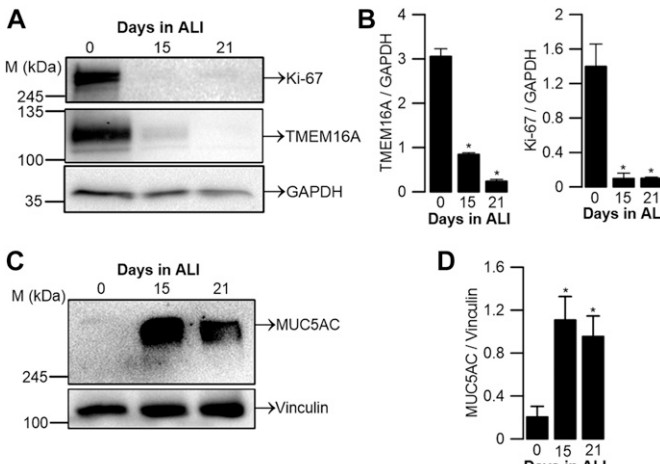

**Figure 4. TMEM16A and Ki-67 protein expression during differentiation of primary HBE cells.**
**(A)** WB of endogenous TMEM16A and Ki-67 proteins during differentiation of primary HBE cells. GAPDH was used as loading control. **(B)** Quantification by densitometry of total TMEM16A and Ki-67 expression detected by WB and normalized to the loading control shown as mean ± SEM (n = 3). **(C)** WB of endogenous MUC5AC during differentiation of primary HBE cells. Vinculin was used as loading control. **(D)** Quantification by densitometry of MUC5AC detected by WB and normalized to the loading control shown as mean ± SEM (n = 3). Asterisks indicate significant difference compared with day 0 (*P*-value < 0.05, unpaired *t* test).
Source data are available for this figure.

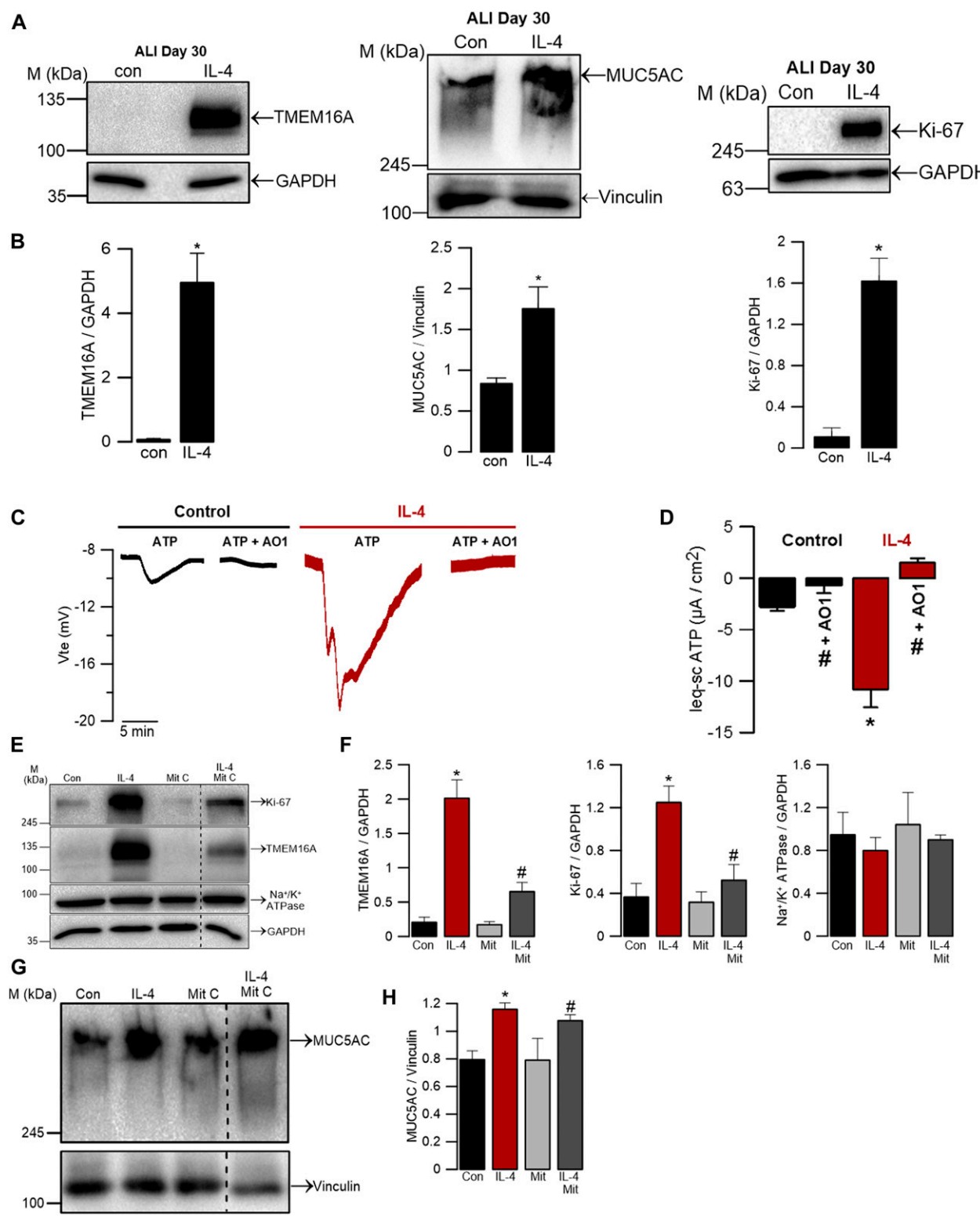

**Figure 5. TMEM16A and MUC5AC up-regulation induced by IL-4 are associated with cell proliferation.**
**(A)** WB indicating up-regulation of TMEM16A, MUC5AC, and Ki-67 by stimulation with 5 ng/ml IL-4 for 48 h in BCi-NS1.1 cells. GAPDH was used as a loading control for TMEM16A and Ki-67 WB. Vinculin was used as a loading control for MUC5AC WB. **(B)** Quantification by densitometry of WB for total TMEM16A, MUC5AC, and Ki-67 expression normalized to the loading control, data shown as mean ± SEM (n = 3). **(C)** Original Ussing chamber tracings ± IL-4, obtained for ATP-induced Cl⁻ currents (100 μM) in the presence of the epithelial Na⁺ channel (ENaC) inhibitor, amiloride (30 μM). Reduction of the ATP-activated Cl⁻ currents was observed under the CaCC-AO1 TMEM16A inhibitor (30 μM). **(D)** Summary of Isc-eq ATP currents in the presence or absence of IL-4. Values are mean ± SEM (n = 3–7). Asterisk indicates significant difference

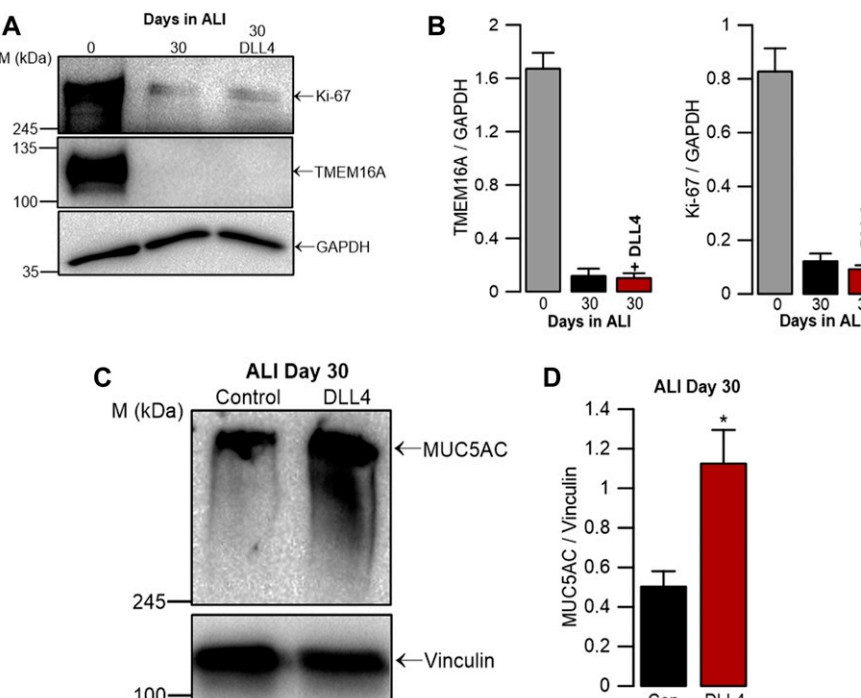

**Figure 6. TMEM16A is not essential for mucus production in BCi-NS1.1 cells.**
**(A)** WB of endogenous TMEM16A and Ki-67 expression in the presence or absence of 400 ng/μl DLL4 for 30 d. Samples from undifferentiated cells were used as a positive control for TMEM16A and Ki-67 expression. GAPDH was used as loading control. **(B)** Quantification by densitometry of total TMEM16A and Ki-67 expression detected by WB and normalized to the loading control shown as mean ± SEM (n = 3). **(C)** WB of endogenous MUC5AC in the presence or absence of 400 ng/μl DLL4. Vinculin was used as a loading control. **(D)** Quantification by densitometry of MUC5AC expression detected by WB and normalized to the loading control shown mean ± SEM (n = 3). Asterisks indicate significant difference compared with control (*P*-value < 0.05, unpaired *t* test).
Source data are available for this figure.

of a $Cl^-/HCO_3^-$ channel, CFTR (Kreda et al, 2012). In this regard, the $Ca^{2+}$-activated $Cl^-$ channel TMEM16A has for long been considered an attractive alternative therapeutic target for these individuals (Verkman & Galietta, 2009; Mall & Galietta, 2015; Li et al, 2017). Pharmacological activation of TMEM16A would, thus, be expected to compensate for the absence of defective CFTR by improving airway hydration through an increase in ASL height (Haq et al, 2016) and mucin solubilization through TMEM16A-dependent $HCO_3^-$ secretion (Jung et al, 2013; Gorrieri et al, 2016).

Nevertheless, recent reports suggest that not activation but rather inhibition of this channel could improve the CF phenotype (Lin et al, 2015; Kunzelmann et al, 2019; Benedetto et al, 2019) by decreasing mucus production and/or secretion (Huang et al, 2012; Lin et al, 2015; Qin et al, 2016; Kondo et al, 2017). However, these studies are only based on the observation that TMEM16A and mucus/MUC5AC are concomitantly up-regulated during inflammation and/or are carried out in animal models, namely, mice (Huang et al, 2012; Lin et al, 2015; Benedetto et al, 2019), that do not mimic the CF airway disease (Huang et al, 2012; Lin et al, 2015; Benedetto et al, 2019).

The main goal of this study was to examine whether TMEM16A drives mucus production (MUC5AC) or if it is just up-regulated by the same stimuli that trigger mucus to contribute to better airway hydration. These findings will contribute to broaden our knowledge

of the role that TMEM16A plays in the airways, with an impact on how it may be used as a drug target for CF and also for other airway obstructive pulmonary diseases (Sondo et al, 2014; Brett, 2015; Sala-Rabanal et al, 2015).

The experiments in this study were performed using a recently described human multipotent airway basal cell line (BCi-NS1.1) isolated from a bronchial brushing of a healthy nonsmoker subject (Walters et al, 2013). This cell line is particularly relevant for the current study because it retains the capacity to multi-differentiate into all airway epithelial cell types (ciliated, goblet, and club), thus being a good model to represent the cellular diversity that characterizes the human airway epithelium (Tam et al, 2011, Fig S1). We have first examined the relationship between TMEM16A and MUC5AC during differentiation of BCi-NS1.1 cells and then we challenged these cells with a pro-inflammatory stimulus, namely, the Th2-cytokine IL-4, and finally we have induced GCM mediated by the Notch1 activator DLL4. Importantly, as this is a novel cell line in the field, crucial experiments were also validated in primary cultures of HBE cells collected from lungs of healthy donors. As results obtained in BCi-NS1.1 and primary HBE cells are consistent (Fig 4), we conclude that this cell line is a good model to recapitulate human airway physiology.

Our results show that cell proliferation is the driver for TMEM16A up-regulation during GCH through several lines of evidence. First,

compared with control; cardinal indicates significant difference compared with ATP alone (*P*-value < 0.05, unpaired *t* test). **(E)** WB showing down-regulation of TMEM16A and Ki-67 expression by treatment with the proliferation blocker mitomycin C (1 μg/ml) and IL-4 (5 ng/μl) for 48 h. $Na^+/K^+$ ATPase (~100 kD)—another protein expressed in the membrane—was detected as a control. GAPDH was used as loading control. Dashed line indicates lanes run on the same gel but noncontiguous. **(F)** Quantification by densitometry of total TMEM16A, Ki-67, and $Na^+/K^+$ ATPase expression detected by WB and normalized to the loading control, shown as mean ± SEM (n = 5–7). **(G)** WB showing MUC5AC levels upon treatment with mitomycin C (1 μg/ml) ± IL-4 (5 ng/ml) for 48 h. Vinculin was used as loading control. Dashed line indicates lanes run on the same gel but noncontiguous. **(H)** Quantification by densitometry of MUC5AC detected by WB and normalized to the loading control, shown as mean ± SEM (n = 4–5). Asterisks indicate significant difference compared with control; cardinals indicate significant difference compared with treatment with IL-4 (*P*-value < 0.05, unpaired *t* test).
Source data are available for this figure.

**A**

DMSO Ani9

0.5 h

4 h

**B**

**Figure 7.** **TMEM16A contributes for fluid secretion in differentiated BCi-NS1.1 cells.**
**(A)** Representative confocal images of ASL labelled with FITC-Dextran obtained 0.5 and 4 h after apical exposure to DMSO or 10 μm Ani9. **(B)** Quantification of overtime (0.5, 1, 2, 3, and 4 h) ASL height measurements shown as mean ± SEM (n = 7–9). Asterisks indicate significant difference compared with 0.5 h (*P*-value < 0.05, unpaired *t* test).

TMEM16A and MUC5AC are inversely correlated as human airway epithelial cells differentiate under control conditions; whereas TMEM16A expression levels are high in non-differentiated (basal) cells and almost undetectable (by WB) in differentiated cells, MUC5AC is very low in the former and abundantly expressed in the latter (Fig 1). The fact that TMEM16A expression is high in basal cells is not totally surprising, given its inducibility by Th2 cytokines (e.g., IL-4 and IL-13). Indeed, these cytokines promote proliferation of basal cells which will then differentiate into goblet cells because of activation of the Notch signaling pathway (Williams et al, 2006). Moreover, those results already suggest that mucus production does not require high levels of TMEM16A. Interestingly, even though TMEM16A protein expression was almost undetectable by WB, immunofluorescence showed an apical staining of TMEM16A in differentiated cells (Fig 2). This is consistent with the pattern that can be found in human bronchi, in which TMEM16A staining in the surface epithelium is apical but at very low levels (Ousingsawat et al, 2009; Caci et al, 2015; Lérias et al, 2018).

Second, our data show that TMEM16A is positively correlated with the proliferation marker Ki-67. This occurs during differentiation under control conditions: fully differentiated cells which do not proliferate also have low levels of TMEM16A (Figs 3A–C and 1A–C). But strikingly, this correlation also occurs when a fully differentiated cell monolayer is subjected to wound healing: the two proteins are up-regulated (Fig 3D and E). This result is in agreement with previous reports in cancer research that link TMEM16A with cell proliferation, migration, and metastasis (Ayoub et al, 2010; Duvvuri et al, 2012; Ruiz et al, 2012; Jia et al, 2015). In addition, another study has shown that knockdown of TMEM16A in primary cultures of human airway epithelial cells of CF patients impairs wound closure (Ruffin et al, 2013).

Third, our data show that induction of TMEM16A by IL-4 is dependent on cell proliferation, whereas mucus hyperproduction is not. Indeed, our data show that when differentiated BCi-NS1.1 cells were exposed to this cytokine, both TMEM16A and MUC5AC protein expression were up-regulated as others reported (Scudieri et al, 2012; Gorrieri et al, 2016). This finding was also corroborated by functional data, as IL-4 also enhanced the TMEM16A-mediated ATP-induced current in Ussing chamber recordings (Fig 5C and D). In fact, IL-4 was reported to modulate the ion transport in the HBE, increasing Cl⁻ secretion and decreasing Na⁺ absorption, thus promoting airway hydration and mucus clearance (Galietta et al, 2002). Remarkably, we show that incubation with IL-4 induced proliferation of fully differentiated cells (Fig 5A and B, right panels), confirming the hypothesis that TMEM16A is up-regulated by IL-4 because of cell proliferation. Indeed, when proliferation is arrested

by mitomycin C, TMEM16A expression induced by IL-4 is significantly reduced (Fig 5E and F), despite MUC5AC still being up-regulated (Fig 5G and H), and thus independent of cell proliferation.

Finally, our data also show that when mucus production is induced by GCM (through modulation of the Notch signalling pathway with the Notch1 activator DLL4), MUC5AC production is significantly increased, but neither TMEM16A nor Ki-67 are up-regulated (Fig 6).

In conclusion, all data presented here clearly show that TMEM16A is only present at high levels in proliferating cells in situations like GCH (Fig 8).

Because these data seem to contradict the conclusion of several studies in the literature, we attempt to conciliate our data with those reports (Huang et al, 2012; Lin et al, 2015; Kondo et al, 2017; Benedetto et al, 2019). It is of relevance to note that mucus synthesis (production) and release (secretion) are controlled by different signalling pathways and that the respective regulators do not entirely correlate in the different airway obstructive pulmonary diseases (Rogers & Barnes, 2006; Fahy & Dickey, 2010). Moreover, the conclusion by previous studies that TMEM16A inhibition improves mucus hypersecretion is strongly based on the use of nonspecific inhibitors (Huang et al, 2012; Lin et al, 2015; Qin et al, 2016; Kondo et al, 2017; Benedetto et al, 2019), which were already shown to act on other chloride channels/transporters, including CFTR (Scott-Ward et al, 2004; Dienna et al, 2007; Benedetto et al, 2017) and other members of the TMEM16 family (Namkung et al, 2011; Wanitchakool et al,

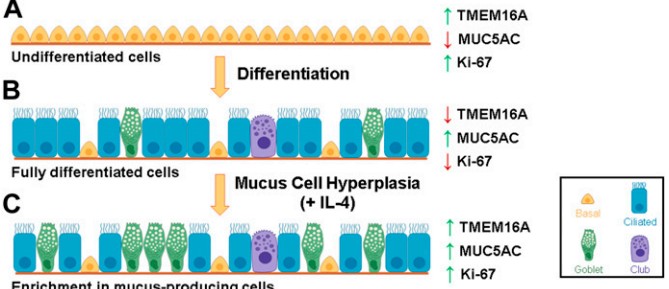

**Figure 8.** **Relationship between TMEM16A, MUC5AC, and Ki-67 during differentiation and GCH.**
**(A)** Undifferentiated basal cells are proliferating and, thus, have high levels of TMEM16A and Ki-67. On the contrary, because of the absence of goblet cells, MUC5AC is not expressed. **(B)** When cells are differentiated, proliferation stops (low TMEM16A and Ki-67) and mucus production is high, mostly secreted from goblet cells. **(C)** Induction of GCH by pro-inflammatory cytokines such as IL-4/13 switches the human airway epithelium from a non-proliferative to a proliferative state, inducing TMEM16A and Ki-67 expression. As a consequence of inflammation, basal cells proliferate and differentiate into mucus-producing cells, increasing MUC5AC levels.

2014; Sirianant et al, 2016) and many, such as niclosamide, were also shown to inhibit cell proliferation (Mazzone et al, 2012; Han et al, 2019).

Other studies have shown that the signaling pathways behind TMEM16A up-regulation induced by IL-4 binding to its membrane receptors activate the transcription factor STAT6, which will then bind to the TMEM16A promoter (Mazzone et al, 2015). Interestingly and consistently, knockdown of STAT6 also impairs cell proliferation (Salguero-Aranda et al, 2019).

Importantly, none of these studies addressed the impact of TMEM16A inhibition on ASL height. In fact, mice lacking TMEM16A exhibit a CF-like phenotype (Ousingsawat et al, 2009; Rock et al, 2009), suggesting that this protein is essential for chloride secretion (Gianotti et al, 2016) and to maintain a proper ASL thickness in mouse airways. Data in the current study also clearly show that TMEM16A inhibition results in a significant decrease in the ASL height in human airway epithelial cells (Fig 7).

Chloride secretion is essential in the airway epithelium to maintain the ASL with a proper thickness, allowing MCT (Haq et al, 2016). Just like CFTR, TMEM16A is expressed at the apical membrane of differentiated airway epithelial cells (Huang et al, 2009; Scudieri et al, 2012), contributing to the alternative route for $Cl^-$ secretion in individuals with CF (Sondo et al, 2014). Interestingly, two recent reports show that TMEM16A is required for CFTR expression and activity (Benedetto et al, 2017, 2019), suggesting that inhibition of TMEM16A would compromise airway hydration. In fact, we show that by blocking TMEM16A with a specific inhibitor (Ani9), the ASL height is significantly reduced, demonstrating that inhibition of this protein would further dehydrate the airways of CF patients.

Taking together all data in this study, we propose a novel model for the mechanism of the relationship between TMEM16A, cell proliferation, and mucus production (Fig 8). In highly proliferating (undifferentiated) basal cells (Fig 8A), TMEM16A expression levels are high. However, as basal cells differentiate (stopping proliferation, Fig 8B) TMEM16A expression levels decrease and its localization becomes specifically apical, whereas MUC5AC levels increase because of appearance of differentiated mucus-producing cells (Fig 8B). Whenever there is inflammation associated to cell proliferation (GCH), MUC5AC production is increased because of enrichment in mucus-producing cells, whereas concomitantly TMEM16A is also up-regulated because of cell proliferation (Fig 8C). As some TMEM16A inhibitors are also inhibitors of cell proliferation (Mazzone et al, 2012; Han et al, 2019), it is very likely that these molecules are reducing mucin levels by inhibiting GCH, albeit through a TMEM16A-independent mechanism.

Given the data presented here, we conclude that finding TMEM16A activators, that is, stimulators of the channel activity acting specifically in non-proliferating differentiated cells, remains a bona fide goal for drug discovery in CF and likely other chronic obstructive airway diseases.

# Materials and Methods

### Human airway BCi-NS1.1 (basal cell immortalized non-smoker) cells

The BCi-NS1.1 cell line was a kind gift from Professor Ronald G. Cristal (Weil Cornell Medical College, New York, USA). As described

by those authors, basal cells were isolated from bronchial brushing of a healthy nonsmoker subject and immortalized using retrovirus-mediated expression of human telomerase reverse transcriptase (hTERT) (Walters et al, 2013).

BCi-NS1.1 were cultured with Pneumacult-Ex Medium supplemented with Pneumacult-Ex 50X supplement (#05008; STEMCELL Technologies), 96 $\mu$g/ml hydrocortisone (H0888; Sigma-Aldrich), and 1% penicillin–streptomycin (10,000 U/ml) (15140-148; Gibco) in a 37°C, 5% $CO_2$ humidified incubator.

Following expansion, the cells were seeded onto either 6.5- or 12-mm-diameter size Transwell inserts with 0.4 $\mu$m pore polyester membrane (#3470, #3460; Corning Incorporated) at a density of $1.5 \times 10^5$ or $3.0 \times 10^5$, respectively. The Transwell inserts were previously coated with human type IV collagen (C7521; Sigma-Aldrich). Cells were cultured with 1:1 DMEM/F-12 (15-090-CM; Corning Incorporated) supplemented with 5% FBS (Gibco), 1% penicillin–streptomycin, 0.5% amphotericin B (15290-026; Gibco), and 0.1% gentamicin (G1272; Gibco). On the following day, the medium in both chambers was replaced with DMEM/F12 supplemented with 2% Ultroser G (15950-017; Pall Life Sciences), 1% penicillin–streptomycin, 0.5% amphotericin B, and 0.1% gentamicin. ALI was established once cells reached full confluency by aspirating the medium in the apical chamber. The medium was replaced every 2–3 d for 30 d, and polarization was monitored by measurements of the TEER using a chopstick electrode (STX2; World Precision Instruments).

### Induction of mucus production in BCi-NS1.1 cells

To induce mucus production with IL-4, the standard ALI method was used and 5 ng/ml of IL-4 (BMS337; eBioscience) was added at ALI day 28 to the basolateral media for 48 h. To study the effect of mitomycin C on IL-4 stimulation, 1 $\mu$g/ml mitomycin C (10107409001; Roche) was added to the basolateral medium in the presence/absence of 5 ng/ml of IL-4 at ALI day 28. To induce GCM, 400 ng/$\mu$l of DLL4 (10171-H02H-25; Life Technologies) was added to the basolateral medium since ALI day 0, after which the standard ALI method was used. The DLL4 stimulation was maintained until ALI day 30 by replacing it at each media change.

### Primary HBE cells

Human lung tissues from healthy donors were obtained from the pediatrics department (Motol University Hospital, Prague, Czech Republic) after receiving patient's written consent and approval by the hospital ethics committee. Primary HBE cells were isolated as previously described (Fulcher & Randell, 2013). After expansion, the cells were differentiated in human type IV collagen–coated Transwell inserts also as described (Moniz et al, 2013).

### qRT-PCR

Gene-specific products for TMEM16A and MUC5AC were amplified from cDNA samples derived from differentiating BCi-NS1.1 cells, using the Evagreen SsoFast PCR reagent (172-5204; Bio-Rad) according to the manufacturer's instructions. The following primer pairs were used: TMEM16A (fwd: 5′-ACTACCACGAGGATGACAAGC-3′; rev: 5′-CTCTGCA-CAGCACGTTCCA-3′) and MUC5AC (fwd: 5′-CCTGAGGGGACGGTGCTT-3′; rev: 5′-ACGAGGTGCAGTTGGTGC-3′). Levels of expression were normalized

against expression of the housekeeping gene GAPDH (fwd: 5′-ATGGGGAAGGTGAAGGTCG-3′; rev: 5′-GGGGTCATTGATGGCAACAATA-3′) in the same samples. Technical duplicates were used in amplification, melt curves were examined to confirm the amplification of specific products, and negative controls were confirmed to be free of amplification after 40 PCR cycles. Mean relative levels of expression were calculated for the two target genes using the ΔΔCT method, where fold-change = $2^{(-\Delta\Delta CT)}$, using mean levels of expression at 30 d of ALI as the baseline.

## Primary antibodies

The following primary antibodies were used in this study: rabbit monoclonal anti-TMEM16A [SP31] (ab64085; Abcam); mouse monoclonal anti-GAPDH [6C5] (ab8245; Abcam); mouse monoclonal anti-MUC5AC [45M1] (MA1-38223; Invitrogen); mouse monoclonal anti-Vinculin [7F9] (sc-73614; Santa Cruz); rabbit monoclonal anti-Ki-67 [SP6] (ab16667; Abcam); mouse monoclonal anti-ZO-1 [1A12] (33-9100; Invitrogen); rabbit monoclonal anti-p63 [EPR5701] (ab124762; Abcam); rabbit polyclonal anti-CC16 (RD181022220-01; BioVendor); and rabbit polyclonal DNAI1 (HPA021649; Sigma-Aldrich).

## WB

BCi-NS1.1 cells grown on 12-mm Transwell inserts were washed twice with ice-cold PBS and lysed with a buffer containing 1.5% (wt/vol) SDS, 10% (vol/vol) glycerol, 0.5 mM DTT, 31.25 mM Tris (pH 6.8), and protease inhibitor cocktail (11697498001; Roche). DNA was sheared by treatment with (5U) benzonase nuclease (E1014; Sigma-Aldrich). Protein lysates were loaded in an acrylamide gel (4% stacking and 7% resolving) under reducing conditions for electrophoresis. Transfer was performed using a wet-transfer system. Membranes were blocked with 5% (wt/vol) non-fat milk in Phosphate Buffered Saline with Tween and probed with anti-Ki-67 (1:250), anti-p63 (1:1,000), anti-DNAI1 (1:1,000), anti-CC16 (1:500), and anti-GAPDH (1:10,000) antibodies diluted in blocking buffer. Membranes probed for TMEM16A were blocked with 1% (wt/vol) non-fat milk diluted in Tris Buffered Saline with Tween and incubated with anti-TMEM16A (1:500) antibody diluted in blocking buffer. TMEM16A and Ki-67 protein levels were always compared on the same blot membrane, with protein collected from the same Transwell insert on the same seeding date. To detect MUC5AC, the cells were lysed with a non-denaturing lysis buffer containing 1% (wt/vol) Triton X-100, 50 mM Tris–HCl (pH 7.4), 300 mM NaCl, 5 mM EDTA, 0.02% wt/vol sodium azide, 1 mM PMSF (10837091001; Roche), 1 mM $Na_3VO_4$ (S6508; Sigma-Aldrich), and protease inhibitor cocktail. DNA was sheared by treatment with (5U) benzonase nuclease. Protein lysates were loaded in an acrylamide gel (4% stacking and 5% resolving) under nonreducing conditions for electrophoresis. Transfer was performed using a wet-transfer system with transfer buffer containing 0.0375% SDS. The membranes were blocked with 5% non-fat milk diluted in Phosphate Buffered Saline with Tween and probed with an anti-MUC5AC (1:1,000) diluted in blocking buffer. All primary antibodies were incubated overnight at 4°C. On the following, the day membranes were washed and incubated with HRP-conjugated goat antimouse or antirabbit IgG (170-6516, 170-6515; Bio-Rad) secondary antibodies (1:3,000) for 1 h at room temperature.

## Immunofluorescence

BCi-NS1.1 cells on Transwell inserts were rinsed three times with ice-cold PBS and fixed by adding 4% (wt/vol) PFA to both apical and basolateral chambers for 15 min at 4°C. After washing, the cells were permeabilised for 15 min with 0.1% (vol/vol, PBS) Triton X-100 in PBS and blocked for 20 min with 1% (wt/vol, PBS) BSA. Transwell inserts were then incubated overnight at 4°C with primary anti-TMEM16A (1:100), anti-ZO-1 (1:100), and anti-MUC5AC (1:100) antibodies. After incubation, the cells were rinsed three times with PBS and incubated for 1 h at room temperature with a solution containing goat antimouse Alexa 488 (A21202; Invitrogen) or goat antirabbit Alexa 568 (A10042; Invitrogen) secondary antibodies (1:500) and Hoechst 33,342 solution (200 ng/ml, #B2261; Sigma-Aldrich) to stain nuclei. In all experiments, a negative control for the primary antibody was used. Transwell inserts were mounted in glass slides with N-propyl-gallate in glycerol–PBS mounting medium and imaged with a Leica TCS SP8 confocal microscope with the 63× oil immersion lens.

## Image analysis

All confocal images were analysed by the following approach: images are represented as average-intensity projections; brightness and contrast are adjusted (being the same criteria applied for each group of images); background was subtracted (being the same value subtracted for each group of images).

## Wound healing

On ALI day 20, a sterile P100 pipette tip was used to scratch the cell monolayer in three different areas, inducing injury. The unattached cells were removed by washing the apical surface twice with PBS. The cells were then returned to the standard ALI culture, and protein was collected from different Transwell inserts at the indicated time points.

## Ussing chamber measurements

Monolayers of BCi-NS1.1 cells cultured for 30 d in ALI were mounted into a micro-Ussing chamber and analysed under open-circuit conditions at 37°C. Apical and basolateral sides were continuously perfused with ringer solutions containing 30 and 145 mM Cl⁻ concentrations (pH 7.4), respectively. After an equilibrium period, 30 $\mu$M amiloride (A7410; Sigma-Aldrich) was added apically to block ENaC. TMEM16A activity was measured by applying 100 $\mu$M ATP (1852; Sigma-Aldrich) in the presence or absence of the preincubated 30 $\mu$M CaCC-AO1 inhibitor (4877; Tocris). ATP or CaCC-AO1 were always added in the presence of 30 $\mu$M amiloride. Values for transepithelial voltages ($V_{te}$) were referenced to the basal surface of the epithelium. Transepithelial resistance ($R_{te}$) was determined by applying short current pulses (1 s) of 0.5 $\mu$A (5-s period). The equivalent short circuit ($I_{eq-sc}$) was calculated according to Ohm's law ($I_{eq-sc} = V_{te}/R_{te}$).

## Measurements of ASL height

Differentiated BCi-NS1.1 cells were washed with PBS for 30 min before performing experiments to remove excess mucus. ASL was

labelled with FITC conjugated to 70-kD Dextran (46945; Sigma-Aldrich). The cells were apically loaded with 20 $\mu$l of an FITC-Dextran solution (in PBS) containing either DMSO or 10 $\mu$M Ani9 (6076; Tocris) to block TMEM16A. Before imaging, cultures were transferred to Ringer's solution, and 100 $\mu$l perfluorocarbon (PFC, FC-770) (F3556; Sigma-Aldrich) was added apically to avoid evaporation. ASL was imaged using an XZ scan on a Leica TCS SP8 confocal microscope with a 63× water immersion lens and the 488-nm laser. For each experiment, the images were acquired at five different points on the Transwell insert. ASL height was measured using the ImageJ software.

### Statistical analysis

All data presented in this study are presented as mean ± SEM. Statistical comparisons were calculated using an unpaired $t$ test. A $P$-value < 0.05 was considered significant.

# Supplementary Information

# Acknowledgements

This work was supported by the UID/MULTI/04046/2019 centre grant from Fundação para a Ciência e a Tecnologia (FCT), Portugal (to BioISI), FCT/POCTI (PTDC/BIM-MEC/2131/2014) grant "DiffTarget" (to MD Amaral), and CF Trust Strategic Research Centre Award (Ref. SRC 003) "Innovative non-CFTR Approaches for Cystic Fibrosis Therapy (INOVCF)" (to MD Amaral). FB Simões and MC Quaresma are recipients of fellowships from BioSys PhD programme PD/BD/114393/2016 (Refs SFRH/PD/BD/131008/2017 and SFRH/PD/BD/114389/2016, respectively) from FCT (Portugal). The authors thank Ronald G Crystal (Department of Genetic Medicine, Weill Cornell Medical College) for kindly providing the BCi-NS1.1 cell line; Dr Tereza Doušová (Motol University Hospital, Prague, Czech Republic) for providing lungs from healthy donors; Karl Kunzelmann (Institut für Physiologie, Universität Regensburg) for fruitful discussions and Robert Tarran (University of North Carolina at Chapel Hill) for helping to establish ASL measurements. The authors are also grateful to Luís Marques, Aires Duarte, and Hugo Botelho (BioISI) for technical assistance. The authors have no additional financial interests.

### Author Contributions

FB Simões: conceptualization, formal analysis, validation, investigation, visualization, methodology, and writing—original draft.
MC Quaresma: formal analysis and methodology.
LA Clarke: formal analysis and methodology.
IAL Silva: methodology.
I Pankonien: conceptualization and methodology.
V Railean: methodology.
A Kmit: formal analysis and methodology.
MD Amaral: resources, supervision, funding acquisition, investigation, project administration, and writing—review and editing.

### Conflict of Interest Statement

The authors declare that they have no conflict of interest.

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
