## [Reviewer comments · Life Science Alliance]

Life Science Alliance

TMEM16A Chloride Channel Does not Drive Mucus Production

Filipa Simões, Margarida Quaresma, Luka Clarke, Iris Silva, Ines Pankonien, Violeta Railean, Arthur Kmit, and Margarida Amaral

DOI: <https://doi.org/10.26508/lsa.201900462>

Corresponding author(s): Margarida Amaral, University of Lisboa

Review Timeline:	Submission Date:	2019-06-20
	Editorial Decision:	2019-07-09
	Revision Received:	2019-10-02
	Editorial Decision:	2019-10-24
	Revision Received:	2019-10-29
	Accepted:	2019-11-06

Scientific Editor: Andrea Leibfried

Transaction Report:

July 9, 2019

Re: Life Science Alliance manuscript #LSA-2019-00462-T

Prof. Margarida D Amaral
University of Lisboa
Faculty of Sciences, BioISI - Biosystems & Integrative Sciences Institute
Lisboa
Portugal

Dear Dr. Amaral,

Thank you for submitting your manuscript entitled "Role of TMEM16A Channel in Mucus Production: Driver or Passenger?" to Life Science Alliance. The manuscript was assessed by expert reviewers, whose comments are appended to this letter.

As you will see, the reviewers appreciate your analyses. They note, however, some conflicting results between immunofluorescence and western blot analyses and between the electrophysiological results and fluid absorption assay that need to get addressed. We would thus like to invite you to submit a revised version of your manuscript to us, addressing these concerns as well as the other (minor) concerns raised by the reviewers.

Thank you for this interesting contribution to Life Science Alliance. We are looking forward to receiving your revised manuscript.

Sincerely,

B. MANUSCRIPT ORGANIZATION AND FORMATTING:

Reviewer #1 (Comments to the Authors (Required)):

The manuscript by Simoes and coll. is an interesting study on the relationship between TMEM16A chloride channel and MUC5AC expression in airway epithelial cells. Authors show that there is an inverse relationship between expression of the two proteins during differentiation of the cells into a mucociliary epithelium. The two proteins are both upregulated during stimulation with IL-4 but the

two events are independently regulated, with TMEM16A being dependent on proliferation. Actually, block of proliferation with mitomycin C also blocks the upregulation of TMEM16A but not the expression of MUC5AC by IL-4. Furthermore, TMEM16A expression is increased when cell proliferation in a differentiated epithelium is resumed by wound healing. For the first time, authors also show that MUC5AC overexpression can be increased independently from TMEM16A by stimulating the cells with DLL4. Although the study is interesting and presenting novel information, there are some important issues that need to be addressed as indicated in my specific comments.

SPECIFIC COMMENTS

- 1) Figure 2: the immunofluorescence XY images do not show a clear difference in TMEM16A expression between day 0 and day 30, in contrast to western blot data (Figure 1). Actually, there seems to be a significant expression of TMEM16A when the epithelium is differentiated.
- 2) Figure 5: authors report that CaCCinh-A01 blocks the effect of ATP, i.e. activation of TMEM16A. However, authors do not show that a second application of ATP is indeed effective. Actually, the lack of ATP effect in the second application could be in part to receptor desensitization. The authors should comment on the complete lack of ATP-induced currents in epithelia not treated with IL-4. This could be consistent with the low levels of TMEM16A protein shown by western blot but then Ani9 should be ineffective on ASL. Are the authors sure that the lack of ATP effect is not due to low expression of purinergic receptor? Would be ionomycin equally ineffective?
- 3) Figure 7: as stated above, how is it possible that Ani9 affects ASL height by blocking TMEM16A if the expression, and particularly function (Figure 5C), are negligible at that stage of epithelial differentiation?
- 4) Abstract: DLL4 results are quite novel and should be mentioned here.
- 5) Introduction, end of page 3: I think that the role of bicarbonate in favoring mucus release and solvation is not entirely clear. The ability of bicarbonate to facilitate mucus release by calcium chelation is probably hypothetical but not demonstrated. Bicarbonate may act by buffering protons or through a direct mechanism on mucins. This part should be modified and original papers reporting the importance of bicarbonate on mucus should be cited (Garcia et al., J Clin Invest 2009; Gustafsson et al., J Exp Med 2012).
- 6) Methods: a better description of the procedure used to wound the epithelium should be provided. Authors say that a pipette tip was used but was it a single linear scratch?

Reviewer #2 (Comments to the Authors (Required)):

This manuscript clearly shows that TMEM16A expression mirrors cell proliferation in the conditions investigated. Mitomycin C and DLL4 (notch inhibitor) experiments show that MUC5AC may be up-regulated without an increase in TMEM16A, which until this manuscript was an established casual relationship. These findings are important and are appropriate for publication in this journal. However, I am unsure why Ani9 affects ASL height in (presumably) untreated differentiated cultures, especially when Figure 5 shows an absence of functional response to ATP. This manuscript is acceptable for publication, contingent upon resolution of this issue and addressing

other comments.

Major Comments:

Page 3 Paragraph 3:

Although many reviews (including those cited here in the manuscript) proposed HCO₃ as a chelating agent. Chelation (more precisely precipitation) has not been directly demonstrated. Carbonate (pKa 10.3) forms precipitates with divalent cations, however at physiological pH values carbonate presence is likely nominal. A more acceptable way to say this is "by promoting calcium exchange from the mucin-containing granules".

Page 10 Paragraph 1:

Last sentence: This is not necessarily true at this point of the manuscript. It is just an inverse correlation (one example scenario could be MUC5AC, once expressed, decreases TMEM16A expression). Please correct. At page 12 end of paragraph 2 ("These data further support.."), this is the first time in the manuscript that the data suggest that TMEM16a and MUC5AC are independent.

Figure 1G

The manuscript would improve by showing TER from supplemental on a right y-axis superimposed on the current panel. The supplemental figure can stay as is.

Figure 3

In legend, please clearly state that TMEM16A summary data is from Figure 1. Also, please clearly state whether Ki-67 was detected from a stripped blot, time-matched cultures performed at the same seeding date, or time-matched cultures performed at a different seeding date. These suggestions do not change interpretations or validity of findings but provide experimental clarity.

Figure 5

According to Methods, Panel D is not I_{sc}, change to I_{eq-sc} on y-axis label.

For the middle panel Ussing chamber experiment, it would be useful to show that ordering of ATP+CaCC-A01 vs. ATP did not matter for these cells (such as performed in Fig4 of Scudieri 2012 J Physiol). It is possible that 10min ATP exposure, with only a short recovery time, may not allow for calcium to re-establish equilibrium for the ATP+CaCC-A01 pulse. Alternatively, an explanation of why it is likely that complete calcium response is intact for the second ATP application would also be acceptable.

Further, if TMEM16A upregulation is in a basal cell, then why does this increase transepithelial Cl secretion?

Figure 7 (major concern)

It is unclear why Ani9 affects ASL height in these experiments. Figure 5C shows no functional TMEM16A response in differentiated cultures. Further, it is unclear how the drugs are added to the apical surface (specify volume and vehicle in methods). If these are IL-4 treated cultures, then it is not clearly stated. The final sentence is complicated and may be more clearer using alternative wording such as: Thus inhibiting TMEM16A causes significant airway dehydration by reducing fluid secretion and TMEM16A potentiation remains a good target for hydrating CF airway.

Minor Edits:

Page 3 Paragraph 1:

"It is a gel formed" ... Define it

The more abundant mucins change to "the main" or "the most abundant"

Final sentence sounds like MUC5AC and MUC5B are exclusively localized, however MUC5B can also be found (although more rarely) in goblet cells (see Bonser & Erle 2017 review)

Page 6 Paragraph 3:

DMEM/F12 should read DMEM/F-12 also state 1:1 DMEM/F-12 when first mentioned to save the reader from having to look up the product for ratio.

State acceptable minimum TEER used for this study.

Page 9 Ussing Chamber Measurements:

Mid-section states all solutions were prepared with 30uM amiloride, however, the epithelia is equilibrated without amiloride. This is confusing. A clearer way of saying this is "ATP or CaCC-A01 was always added in the presence of amiloride".

Figure 1 legend:

Panel B: Is the densitometry for TMEM16A performed on glycosylated or non-glycosylated band? Both obviously decrease, but state which band in legend for precision.

Figure 2

Scale bar text is not readable. Can omit text as scale bar is clearly stated in legend. Also figure 2A Day 30 Merged (lower right image) is missing its scale bar.

Page 15 Paragraph 2 (Our results...)

The inverse correlation between TMEM16A and MUC5AC is not strong evidence that these proteins are unrelated to each other. These data could suggest the authors conclusion and their other data certainly reinforce that they are unrelated.

Response to reviewers' comments**Reviewer #1** (Comments to the Authors (Required)):

The manuscript by Simoes and coll. is an interesting study on the relationship between TMEM16A chloride channel and MUC5AC expression in airway epithelial cells. Authors show that there is an inverse relationship between expression of the two proteins during differentiation of the cells into a mucociliary epithelium. The two proteins are both upregulated during stimulation with IL-4 but the two events are independently regulated, with TMEM16A being dependent on proliferation. Actually, block of proliferation with mitomycin C also blocks the upregulation of TMEM16A but not the expression of MUC5AC by IL-4. Furthermore, TMEM16A expression is increased when cell proliferation in a differentiated epithelium is resumed by wound healing. For the first time, authors also show that MUC5AC overexpression can be increased independently from TMEM16A by stimulating the cells with DLL4. Although the study is interesting and presenting novel information, there are some important issues that need to be addressed as indicated in my specific comments.

Our response:

We greatly appreciate the Reviewer's positive view of the work.

SPECIFIC COMMENTS

1) *Figure 2: the immunofluorescence XY images do not show a clear difference in TMEM16A expression between day 0 and day 30, in contrast to western blot data (Figure 1). Actually, there seems to be a significant expression of TMEM16A when the epithelium is differentiated.*

Our response:

The immunofluorescence images in the previous Figure 2 were represented as maximum intensity projections of all the Z-stacks acquired by the confocal microscope. All confocal images in the revised Figure 2 were now modified according to a new analysis, which includes the following changes:

- Images are represented as average intensity projections;
- Brightness and contrast are adjusted (being the same criteria applied for each group of images);
- Background was subtracted (being the same value subtracted for each group of images),

This image analysis approach is now described in the Methods section (p.9, 1st parag).

2) *Figure 5: authors report that CaCCinh-A01 blocks the effect of ATP, i.e. activation of TMEM16A. However, authors do not show that a second application of ATP is indeed effective. Actually, the lack of ATP effect in the second application could be in part to receptor desensitization. The authors should comment on the complete lack of ATP-induced currents in epithelia not treated with IL-4. This could be consistent with the low levels of TMEM16A protein shown by western blot but then Ani9 should be ineffective on ASL. Are the authors sure that the lack of ATP effect is not due to low expression of purinergic receptor? Would be ionomycin equally ineffective?*

Our response:

Novel Ussing chamber experiments were performed and an ATP-induced chloride current was detected both in the presence and in the absence of CaCC-AO1 inhibitor in control cells.

Nevertheless, pre-incubation with the AO1 inhibitor significantly decreased the ATP-induced current (Fig.5 C, D). Moreover, after removing the ATP + CaCC-AO1, a third application of ATP was still effective (see Figure 1 for inspection below which is now included in revised Figure S2). Experiments in previous Fig.5 had been performed after stimulation and inhibition of the CFTR channel, which might have "masked" the occurrence of a subsequent ATP-induced current, especially when TMEM16A expression levels are low (Fig.1 A, B, C).

Figure 1 for inspection: Original Ussing chamber tracing from differentiated control cells obtained for ATP-induced Cl^- currents ($100 \mu\text{M}$) in the presence of the epithelial Na^+ channel (ENaC) inhibitor, amiloride ($30 \mu\text{M}$) +/- CaCC-AO1 TMEM16A inhibitor ($30 \mu\text{M}$).

3) *Figure 7: as stated above, how is it possible that Ani9 affects ASL height by blocking TMEM16A if the expression, and particularly function (Figure 5C), are negligible at that stage of epithelial differentiation?*

Our response:

Incubation of untreated differentiated BCI-NS1.1 cells with Ani9 leads to a decrease in the ASL height by blocking TMEM16A (Fig.7). Even though TMEM16A expression is low in differentiated BCI-NS1.1 cells, the levels present at the apical membrane of some cells (Fig.2 A, B) and detected by Western blot (Fig.1 A, B) seem to be sufficient to induce a transepithelial ATP-induced chloride current in the Ussing chamber (revised Fig.5) and to significantly contribute for fluid secretion (Fig.7). In fact, TMEM16A expression in untreated differentiated BCI-NS1.1 cells is comparable to that in primary human bronchial epithelial cells as shown in the present study (Fig.4 A, B) and also previously by Caci *et al*, *Plos One* 2015. Moreover, our results are also consistent with the pattern that can be found in healthy human bronchi by several groups, in which TMEM16A staining is detected at the apical surface epithelium albeit at very low levels (Caci *et. al*, *Plos One* 2015; Lérias *et al*, *BBA Molecular Cell Research* 2018; Ousingsawat *et. al*, *JBC* 2009).

4) *Abstract: DLL4 results are quite novel and should be mentioned here.*

Our response:

This information was added to the abstract: "*Consistently, promotion of differentiation into mucus-producing cells with the Notch1 ligand DLL4 did not induce any change in TMEM16A and Ki-67 expression, despite the significant increase in MUC5AC levels.*"

5) *Introduction, end of page 3: I think that the role of bicarbonate in favouring mucus release and solvation is not entirely clear. The ability of bicarbonate to facilitate mucus release by*

calcium chelation is probably hypothetical but not demonstrated. Bicarbonate may act by buffering protons or through a direct mechanism on mucins. This part should be modified and original papers reporting the importance of bicarbonate on mucus should be cited (Garcia et al, J Clin Invest 2009; Gustafsson et al, J Exp Med 2012).

Our response:

The text was changed to (p.3, last parag): "*This is further exacerbated because of CFTR permeability to HCO_3^- , which is essential in the extracellular space for proper mucus release, probably by chelating the Ca^{2+} and H^+ present in the mucin-containing intracellular granules, thus facilitating mucin expansion.*" Original papers were also cited.

6) Methods: a better description of the procedure used to wound the epithelium should be provided. Authors say that a pipette tip was used but was it a single linear scratch

Our response:

The transwells were scratched with a pipette tip in three different areas (three different scratches). This information was added to the manuscript (p.9, 2nd parag): "*On ALI day 20, a sterile P100 pipette tip was used to scratch the cell monolayer in three different areas, inducing injury*".

Reviewer #2 (Comments to the Authors (Required)):

This manuscript clearly shows that TMEM16A expression mirrors cell proliferation in the conditions investigated. Mitomycin C and DLL4 (notch inhibitor) experiments show that MUC5AC may be up-regulated without an increase in TMEM16A, which until this manuscript was an established casual relationship. These findings are important and are appropriate for publication in this journal.

Our response:

We greatly appreciate the Reviewer's positive view of the work.

However, I am unsure why Ani9 affects ASL height in (presumably) untreated differentiated cultures, especially when Figure 5 shows an absence of functional response to ATP. This manuscript is acceptable for publication, contingent upon resolution of this issue and addressing other comments.

Our response:

Novel Ussing chamber experiments were performed and an ATP-induced current sensitive to CaCC-AO1 was detected in control cells and found to be significantly lower in comparison to that in cells treated with IL-4 (see revised Fig.5 C, D). Moreover, a third application of ATP (after removing ATP + CaCC-AO1) was still effective (see Figure 1 for inspection below, which is now included in revised Figure S2).

Experiments in previous Fig.5 had been performed after stimulation and inhibition of the CFTR channel, which might have "masked" the occurrence of a subsequent ATP-induced current, especially when TMEM16A expression levels are low (Fig.1 A, B, C).

Accordingly, incubation of untreated differentiated BCI-NS1.1 cells with Ani9 leads to a decrease in the ASL height and it is fair to conclude that this is due to blocking of TMEM16A (Fig.7). Even though TMEM16A expression is low in differentiated BCI-NS1.1 cells, the levels present at the apical membrane of some cells (Fig.2 A, B) seem to be sufficient to induce a transepithelial chloride secretion in the using chamber (revised Fig.5 C, D) and to significantly contribute for fluid secretion (Fig.7).

Major Comments:

Page 3 Paragraph 3: Although many reviews (including those cited here in the manuscript) proposed HCO₃ as a chelating agent. Chelation (more precisely precipitation) has not been directly demonstrated. Carbonate (pKa 10.3) forms precipitates with divalent cations, however at physiological pH values carbonate presence is likely nominal. A more acceptable way to say this is "by promoting calcium exchange from the mucin-containing granules".

Our response:

The text was modified according to suggestion and original papers regarding the role of bicarbonate in mucus expansion were cited (Garcia *et al*, *J Clin Invest* 2009; Gustafsson *et al*, *J Exp Med* 2012) (p.3, last parag): "*This is further exacerbated because of CFTR permeability to HCO₃⁻, which is essential in the extracellular space for proper mucus release, probably by promoting Ca²⁺ and H⁺ exchange from the mucin-containing intracellular granules, thus facilitating mucin expansion (14,15)*".

Page 10 Paragraph 1: Last sentence: This is not necessarily true at this point of the manuscript. It is just an inverse correlation (one example scenario could be MUC5AC, once expressed,

decreases TMEM16A expression). Please correct. At page 12 end of paragraph 2 ("These data further support..), this is the first time in the manuscript that the data suggest that TMEM16a and MUC5AC are independent.

Our response:

The text was modified (p.11, 1st parag): "These data show an inverse correlation between TMEM16A and MUC5AC expression during differentiation"

Figure 1G The manuscript would improve by showing TER from supplemental on a right y-axis superimposed on the current panel. The supplemental figure can stay as is.

Our response:

Figures 1G and 3C were modified as recommended and the respective legends adjusted.

Figure 3 In legend, please clearly state that TMEM16A summary data is from Figure 1.

Our response:

The legend was modified as recommended.

Also, please clearly state whether Ki-67 was detected from a stripped blot, time-matched cultures performed at the same seeding date, or time-matched cultures performed at a different seeding date. These suggestions do not change interpretations or validity of findings but provide experimental clarity.

Our response:

TMEM16A and Ki-67 were detected on the same Western blot, without stripping the membrane, as the molecular weights allowed it (TMEM16A ~100 kDa; Ki-67 ~ 245 kDa). This information was already included in the Methods section (Western Blot) (p.8, 1st parag): "TMEM16A and Ki-67 protein levels were always compared on the same blot membrane, with protein collected from the same transwell on the same seeding date".

Figure 5 According to Methods, Panel D is not I_{sc}, change to I_{eq-sc} on y-axis label.

Our response:

The Y-axis label in Fig.5 was adjusted as indicated.

For the middle panel Ussing chamber experiment, it would be useful to show that ordering of ATP+CaCC-A01 vs. ATP did not matter for these cells (such as performed in Fig4 of Scudieri 2012 J Physiol). It is possible that 10min ATP exposure, with only a short recovery time, may not allow for calcium to re-establish equilibrium for the ATP+CaCC-A01 pulse. Alternatively, an explanation of why it is likely that complete calcium response is intact for the second ATP application would also be acceptable.

Our response:

Ussing chamber experiments were repeated and an ATP-induced chloride current was detected in the presence and in the absence of CaCC-AO1 in control cells. Pre-incubation with the inhibitor decreased significantly the ATP-induced current (Fig.5 C, D). Moreover, after removing the ATP + CaCC-AO1, a third application of ATP was still effective (see Figure 1 for inspection below, which is now included in revised Figure S2), showing that the wash-out period (20 min) was enough to re-establish the calcium intracellular levels.

Figure 1 for inspection: Original Ussing chamber tracing from differentiated control cells obtained for ATP-induced Cl^- currents ($100 \mu\text{M}$) in the presence of the epithelial Na^+ channel (ENaC) inhibitor, amiloride ($30 \mu\text{M}$) +/- CaCC-AO1 TMEM16A inhibitor ($30 \mu\text{M}$).

Further, if TMEM16A upregulation is in a basal cell, then why does this increase transepithelial Cl^- secretion?

Our response:

Basal cells are responsible to regenerate the airway epithelium into different proportions of basal, ciliated and secretory cells, according to signalling pathways. Airway inflammation induced by Th2 cytokines (IL-4) promotes proliferation of basal cells which will then differentiate into goblet cells due to activation of the Notch signalling pathway (Williams *et al*, *Am J Respir Cell Mol Biol* 2006). The increase in transepithelial chloride secretion induced by IL-4 is likely a result from TMEM16A expression in goblet cells that were generated from proliferating basal cells. Some of these considerations have now been incorporated into the Discussion (see p.16, 2nd parag).

Figure 7 (major concern) It is unclear why Ani9 affects ASL height in these experiments. Figure 5C shows no functional TMEM16A response in differentiated cultures. Further, it is unclear how the drugs are added to the apical surface (specify volume and vehicle in methods). If these are IL-4 treated cultures, then it is not clearly stated.

Our response:

Incubation of untreated differentiated BCI-NS1.1 cells with Ani9 leads to a decrease in the ASL height by blocking TMEM16A (Fig.7). Even though TMEM16A expression is low in differentiated BCI-NS1.1 cells (Fig. 1 A, B), the levels present at the apical membrane of some cells (Fig.2 A, B) seem to be sufficient to induce a transepithelial chloride secretion in the Ussing chamber (revised Fig.5 C, D) and to significantly contribute to fluid secretion (Fig.7). In fact, TMEM16A expression in untreated differentiated BCI-NS1.1 cells is comparable to that in primary human bronchial epithelial cells as shown both in Fig.4 A, B and by others [Caci *et al*, *Plos One* 2015]. Moreover, our results are also consistent with the TMEM16A expression pattern found in healthy human native bronchi, in which TMEM16A is detected at the apical surface epithelium albeit at very low levels (Caci *et al*, *Plos One* 2015; Lérias *et al*, *BBA Molecular Cell Research* 2018; Ousingsawat *et al*, *JBC* 2009).

Information regarding how Ani9 and vehicle were added to untreated cultures was added to the text in the Methods (see p.9, last parag): "ASL was labelled with fluorescein isothiocyanate (FITC) conjugated to 70 kDa Dextran (Sigma Aldrich, 46945). Cells were apically loaded with 20

μL of a FITC-Dextran solution (in PBS) containing either DMSO or 10 μM Ani9 (Tocris, 6076) to block TMEM16A."

The final sentence is complicated and may be more clearer using alternative wording such as: Thus inhibiting TMEM16A causes significant airway dehydration by reducing fluid secretion and TMEM16A potentiation remains a good target for hydrating CF airway.

Our response:

The text was modified as suggested (see p.14, last parag).

Minor Edits:

Page 3 Paragraph 1: "It is a gel formed"... Define it

Our response:

The text was modified as follows (see p.3, 1st parag): "*Mucus is a gel formed by 97% water and 3% solids (mucins, non-mucin proteins, ions, lipids and antimicrobial peptides)*"

The more abundant mucins change to "the main" or "the most abundant"

Our response:

The text was modified to (see p.3, 1st parag): "*The main mucins present (...)*".

Final sentence sounds like MUC5AC and MUC5B are exclusively localized, however MUC5B can also be found (although more rarely) in goblet cells (see Bonser & Erle 2017 review)

Our response:

The text was modified as follows (see p.3, 1st parag): "*The main mucins present in human airway mucus are MUC5AC and MUC5B which are mostly secreted from goblet cells at the surface airway epithelium and by submucosal glands, respectively*" and a citation to the Bonser & Erle, *J Clin Med* 2017 review was included (new Ref.5).

Page 6 Paragraph 3: DMEMF12 should read DMEM/F-12 also state 1:1 DMEM/F-12 when first mentioned to save the reader from having to look up the product for ratio. State acceptable minimum TEER used for this study.

Our response:

The text was modified as indicated twice (see p.6, 3rd parag). The TEER used in this study varied considerably during differentiation. Depending on the experiment, different time-points of differentiation were studied. Notwithstanding, a graph of TEER at the different time-points is shown in Fig.S1.

Page 9 Ussing Chamber Measurements: Mid-section states all solutions were prepared with 30uM amiloride, however, the epithelia is equilibrated without amiloride. This is confusing. A clearer way of saying this is "ATP or CaCC-A01 was always added in the presence of amiloride".

Our response:

The text was modified as suggested (see p.9, 3rd parag).

Figure 1 legend: Panel B: Is the densitometry for TMEM16A performed on glycosylated or non-glycosylated band? Both obviously decrease, but state which band in legend for precision.

Our response:

The densitometry quantifications of TMEM16A were always performed with both glycosylated and non-glycosylated bands (i.e., for total protein). This information was added to the legends of all figures showing TMEM16A Western blots (i.e., Figs.1, 3, 4, 5 and 6).

Figure 2 Scale bar text is not readable. Can omit text as scale bar is clearly stated in legend. Also figure 2A Day 30 Merged (lower right image) is missing its scale bar.

Our response:

The text from scale bar was omitted and scale bar length is indicated just in the legend of Fig.2. Also, in Figure 2A-Day 30 Merged (lower right image) now a scale bar has been included.

Page 15 Paragraph 2 (Our results...) The inverse correlation between TMEM16A and MUC5AC is not strong evidence that these proteins are unrelated to each other. These data could suggest the authors conclusion and their other data certainly reinforce that they are unrelated.

Our response:

The text was modified as follows: "Those results already suggest that mucus production does not require high levels of TMEM16A" (see p.16, 2nd parag.).

October 24, 2019

RE: Life Science Alliance Manuscript #LSA-2019-00462-TR

Prof. Margarida D Amaral
University of Lisboa
Faculty of Sciences, BioISI - Biosystems & Integrative Sciences Institute
Lisboa
Portugal

Dear Dr. Amaral,

Thank you for submitting your revised manuscript entitled "Role of TMEM16A Channel in Mucus Production: Driver or Passenger?". We would be happy to publish your paper in Life Science Alliance pending final revisions necessary to meet our formatting guidelines.

As you will see, while reviewer #1 thinks that the extent of TMEM16A inhibition by CaCCinh-A01 could have been better analyzed, overall the reviewers are satisfied with the changes introduced in revision. A few issues need addressing still, though:

- Please provide the source data for Fig 1D, 5A, 5E, S1B
- I would like to suggest a change of title to avoid the impression that the article is a review
- Please add a running title in our system
- Please add a callout to Fig 1G in the manuscript text
- Please add individual panel descriptors to your figure 8 (those mentioned in legend; A-C)
- Please list 10 authors et al in your reference list

A. FINAL FILES:

B. MANUSCRIPT ORGANIZATION AND FORMATTING:

Sincerely,

Reviewer #1 (Comments to the Authors (Required)):

The authors have not entirely resolved the issue raised for Figure 5 (i.e. extent of TMEM16A inhibition by CaCCinh-A01) given that the inhibitor was applied during the second application of ATP, with the possibility of overestimation of inhibition due to purinergic receptor desensitization. Authors now show in Supplementary Figure 2 that a third ATP application (without inhibitor) is effective. However, the peak remains very low, showing no recovery with respect to second stimulus. A simple experiment could have been to show a double stimulation with ATP but without inhibitor on the second stimulation, or, even better, a comparison of single ATP stimulation in the presence and absence of inhibitor.

Despite this issue, the paper conveys the interesting and novel information about the relationship between TMEM16A and cell proliferation. When cell proliferation is stimulated by wounding the epithelium, or treating with IL-4, TMEM16A expression is induced. Importantly, TMEM16A but not MUC5AC upregulation can be blocked by arresting cell proliferation with mitomycin C. Furthermore, treatment with the NOTCH ligand DLL4 induces MUC5AC but not TMEM16A. Such results indicate that MUC5AC expression levels are independent from those of TMEM16A. The data presented in the paper certainly open a new field of investigation.

Reviewer #2 (Comments to the Authors (Required)):

This manuscript teases out the correlation between TMEM16A and MUC5AC. The authors convincingly show that TMEM16A mirrors cell proliferation and is not necessarily required for MUC5AC to be up-regulated. These findings shed new light upon this TMEM16A and MUC5AC relationship. The authors carefully evaluated and addressed reviewers' comments and this manuscript should be accepted for publication.

Response to editorial points and reviewers' comments**Editorial points to be addressed:**

- Please provide the source data for Fig 1D, 5A, 5E, S1B

Our response:

The original data are included as single figures.

- I would like to suggest a change of title to avoid the impression that the article is a review

Our response:

The original title has been changed into: "TMEM16A Chloride Channel Does not Drive Mucus Production".

- Please add a running title in our system

Our response:

A running title has been included: "TMEM16A and Mucus Production".

- Please add a callout to Fig 1G in the manuscript text

Our response:

A callout to Fig 1G has been included in the manuscript text: (p.8, 1st parag, last line)

- Please add individual panel descriptors to your figure 8 (those mentioned in legend; A-C)

Individual panel descriptors to Fig.8 have been included in the figure and in the manuscript text: (p.15, last parag).

- Please list 10 authors et al in your reference list

Our response:

This change in the formatting of references has been introduced (see Reference list).

Response to reviewers' comments**Reviewer #1 (Comments to the Authors (Required)):**

The authors have not entirely resolved the issue raised for Figure 5 (i.e. extent of TMEM16A inhibition by CaCCinh-A01) given that the inhibitor was applied during the second application of ATP, with the possibility of overestimation of inhibition due to purinergic receptor desensitization. Authors now show in Supplementary Figure 2 that a third ATP application (without inhibitor) is effective. However, the peak remains very low, showing no recovery with respect to second stimulus. A simple experiment could have been to show a double stimulation with ATP but without inhibitor on the second stimulation, or, even better, a comparison of single ATP stimulation in the presence and absence of inhibitor.

Despite this issue, the paper conveys the interesting and novel information about the relationship between TMEM16A and cell proliferation. When cell proliferation is stimulated by wounding the epithelium, or treating with IL-4, TMEM16A expression is induced. Importantly, TMEM16A but not MUC5AC upregulation can be blocked by arresting cell proliferation with mitomycin C. Furthermore, treatment with the NOTCH ligand DLL4 induces MUC5AC but not

TMEM16A. Such results indicate that MUC5AC expression levels are independent from those of TMEM16A. The data presented in the paper certainly open a new field of investigation.

Our response:

We thank the Reviewer's suggestion which indeed would have been another way of tackling the issue of TMEM16A inhibition by CaCCinh-A01. We also greatly appreciate the Reviewer's positive view of the work.

Reviewer #2 (Comments to the Authors (Required)):

This manuscript teases out the correlation between TMEM16A and MUC5AC. The authors convincingly show that TMEM16A mirrors cell proliferation and is not necessarily required for MUC5AC to be up-regulated. These findings shed new light upon this TMEM16A and MUC5AC relationship. The authors carefully evaluated and addressed reviewers' comments and this manuscript should be accepted for publication.

Our response:

We greatly appreciate the Reviewer's positive view of the work and recommendation to publish our work.

November 6, 2019

RE: Life Science Alliance Manuscript #LSA-2019-00462-TRR

Prof. Margarida D Amaral
University of Lisboa
Faculty of Sciences, BioISI - Biosystems & Integrative Sciences Institute
Campo Grande, C8 bdg
Lisboa 1749-016
Portugal

Dear Dr. Amaral,

Thank you for submitting your Research Article entitled "TMEM16A Chloride Channel Does not Drive Mucus Production". I appreciate the introduced changes and that you now indicate all splices in the western blots and display blots matching the source data provided. I am also glad that you clarified in your recent correspondence to me the previous mis-matches between source data and the data in the figures. I am thus happy to let you know that your manuscript is now accepted for publication in Life Science Alliance. Congratulations on this interesting work.

*****IMPORTANT:** If you will be unreachable at any time, please provide us with the email address of an alternate author. Failure to respond to routine queries may lead to unavoidable delays in publication.*******

DISTRIBUTION OF MATERIALS:

Again, congratulations on a very nice paper. I hope you found the review process to be constructive

and are pleased with how the manuscript was handled editorially. We look forward to future exciting submissions from your lab.

Sincerely,

Andrea Leibfried, PhD
Executive Editor
Life Science Alliance
Meyerohofstr. 1
69117 Heidelberg, Germany
t +49 6221 8891 502
e a.leibfried@life-science-alliance.org
www.life-science-alliance.org